# Balancing Decidualization, Autophagy, and Cellular Senescence for Reproductive Success in Endometriosis Biology

**DOI:** 10.3390/ijms26189125

**Published:** 2025-09-18

**Authors:** Hiroshi Shigetomi, Miki Nishio, Mai Umetani, Shogo Imanaka, Hiratsugu Hashimoto, Hiroshi Kobayashi

**Affiliations:** 1Department of Obstetrics and Gynecology, Nara Medical University, 840 Shijo-cho, Kashihara 634-8522, Japan; hshige35@gmail.com (H.S.); shogo_0723@naramed-u.ac.jp (S.I.); 2Department of Gynecology and Reproductive Medicine, Aska Ladies Clinic, 3-3-17 Kitatomigaoka-cho, Nara 634-0001, Japan; 3Department of Gynecology and Reproductive Medicine, Ms.Clinic MayOne, 871-1 Shijo-cho, Kashihara 634-0813, Japan; mikin.yaruki10000.boshi@gmail.com (M.N.); mai_umetani@yahoo.co.jp (M.U.); hiratsugu_hashimoto@yahoo.co.jp (H.H.)

**Keywords:** AMPK, decidualization, mTOR, p53, senescence

## Abstract

Endometriosis is a chronic disease characterized by the ectopic presence of endometrial cells that evade apoptosis and survive and proliferate under harsh environmental conditions. It is closely associated with infertility and pregnancy-related complications. This review focuses on the molecular pathophysiology of endometriosis, particularly the disruption of the p53–AMPK–mTOR signaling axis, and highlights the dysregulation of decidualization and cellular senescence, incorporating recent findings in reproductive physiology. A comprehensive literature search was conducted using PubMed and Google Scholar without temporal restrictions. Endometriotic cells adapt to the hostile peritoneal environment through resistance to apoptosis and alterations in autophagy. In the early stages, autophagy activation may promote cell survival; however, as the disease progresses, autophagic activity tends to decline. Aberrant activation of mTOR signaling is implicated in this process, contributing to the suppression of autophagy, impaired decidualization, and promotion of cellular senescence, ultimately facilitating lesion progression and infertility. Indeed, in the eutopic endometrium of patients with endometriosis, progesterone resistance, elevated inflammatory cytokines, and epigenetic abnormalities are known to reduce endometrial receptivity. Moreover, suppression of autophagy leads to excessive cellular senescence and secretion of the senescence-associated secretory phenotype (SASP), thereby interfering with proper decidualization. Maintaining an appropriate balance between decidualization and cellular senescence is essential for reproductive function. Future development of therapeutic strategies targeting these processes is expected to help overcome infertility associated with endometriosis.

## 1. Introduction

Endometriosis is a chronic inflammatory disease that affects approximately 10% of women of reproductive age and is recognized as a major cause of dysmenorrhea and infertility [1]. This condition is characterized by the ectopic presence of endometrial-like tissue outside the uterine cavity—particularly on the peritoneum and ovaries—which undergoes cyclical bleeding and proliferation in response to menstrual hormones [2]. However, the mechanisms by which endometrial cells, which would normally be eliminated by apoptosis and immune surveillance, are able to survive, proliferate, and establish lesions within the peritoneal cavity remain incompletely understood [3]. Studies have revealed that endometrial cells in endometriosis adapt to a microenvironment marked by hypoxia, oxidative stress, excessive inflammatory cytokines, and hormonal imbalance, thereby acquiring resistance to apoptosis and the ability to evade immune responses, which in turn facilitates lesion formation and persistence [2]. Moreover, emerging evidence indicates that cellular homeostatic mechanisms, such as autophagy and senescence, are deeply involved in the onset and progression of endometriosis [4,5]. These intracellular responses are finely regulated by signaling pathways including p53 (tumor protein p53), AMPK (AMP-activated protein kinase), and mTOR (mechanistic target of rapamycin), and disruptions in these pathways are believed to significantly influence cell survival, decidualization, and the establishment of a receptive endometrial environment [6,7].

While the advancement of assisted reproductive technology (ART) has enabled many couples to achieve pregnancy, the success rate of ART remains limited. Traditionally, infertility associated with endometriosis has been attributed primarily to the diminished quality and quantity of oocytes. However, accumulating evidence now suggests that, in addition to ovarian function, the functional maturation of the endometrium is critical for successful pregnancy, and its impairment may represent a key contributor to infertility [8,9]. Recent findings have shown that, during decidualization, a subset of stromal cells exhibits an acute senescence-like phenotype. These senescent cells transiently create a localized inflammatory milieu, which may promote the formation of a receptive endometrium favorable for embryo implantation [10,11]. Thus, the appropriate balance and maintenance of decidualization and cellular senescence within the endometrium are crucial for implantation and pregnancy maintenance, and their disruption is likely to contribute to infertility [12]. We have recently reviewed the molecular mechanisms underlying age-related endometrial dysfunction [6]. Our findings suggest that aging disrupts the balance between decidualization and senescence, leading to the excessive accumulation of senescent cells, chronic inflammation associated with sustained SASP expression, fibrosis, and disturbances in the local immune environment [6]. This pathological senescent milieu has been implicated in implantation failure, early miscarriage, and impaired pregnancy maintenance. Accordingly, qualitative changes in the endometrium are increasingly recognized as significant contributors to reduced fertility potential [12]. These findings suggest that while a moderate level of cellular senescence is beneficial for maintaining endometrial homeostasis, excessive accumulation of senescent cells due to aging may impair reproductive function and contribute to infertility.

In this review article, we aim to provide an overview of the latest molecular mechanisms related to apoptosis evasion, dysregulated autophagy, decidualization, and cellular senescence in endometriosis. Specifically, we examine how abnormalities in the p53/AMPK/mTOR signaling pathways contribute to lesion formation and progression, as well as to reduced fertility potential, with the ultimate goal of advancing our understanding of endometriosis pathophysiology and informing the development of novel therapeutic strategies.

## 2. Pathophysiology of Endometriosis

Ectopically proliferating endometrial tissue undergoes cyclic bleeding in response to hormonal fluctuations, leading to chronic inflammation in the surrounding tissues. This persistent inflammatory state contributes to reduced fertility not only through structural factors such as pelvic adhesions and anatomical distortions but also via functional mechanisms. These include hormonal imbalances, abnormal folliculogenesis, decreased oocyte quality, toxicity to sperm and embryos, impaired tubal motility and embryo transport, and diminished endometrial receptivity [1]. Endometriotic ovarian cysts (chocolate cysts) can compress and damage normal ovarian tissue, potentially reducing ovarian reserve. In addition, surgical treatment of these cysts may cause further damage to ovarian tissue, increasing the risk of a decline in oocyte count [13]. Immune system dysregulation is also thought to play a role in the pathogenesis of endometriosis [14,15]. For example, aberrant activation of peritoneal macrophages and innate immune cells may lead to heightened immune responses against sperm and fertilized eggs, thereby impairing fertilization and implantation. Moreover, a subset of patients with endometriosis exhibits decreased responsiveness to progesterone in the endometrium, a phenomenon known as “progesterone resistance” [9]. This resistance may impair decidualization, a process essential for embryo implantation, thereby contributing to implantation failure [9]. In fact, impaired decidualization in the endometrium is closely associated with implantation failure and is considered one of the underlying causes of infertility. In addition to progesterone resistance, several other factors have been implicated in the impaired fertility associated with endometriosis. These include defective decidualization, oxidative stress, chronic inflammation, abnormalities in metabolic reprogramming, cellular senescence, resistance to apoptosis, and impaired autophagic mechanisms, and epigenetic dysregulation. This article provides an overview of the latest findings regarding the molecular mechanisms underlying each of these factors.

### 2.1. Hormonal Regulation of Decidualization

Estrogen and progesterone are essential for reproductive function and the maintenance of pregnancy, and their coordinated actions establish the normal menstrual cycle and implantation environment [16,17]. During the proliferative phase, estrogen maintains endometrial proliferation and structural integrity through estrogen receptor alpha/beta (ERα/ERβ), while in the secretory phase after ovulation, progesterone induces decidualization via the progesterone receptor (PGR) [18,19]. In addition to the classical genomic pathway mediated by PGR, progesterone exerts non-genomic effects through membrane progesterone receptors (mPR) and the progesterone receptor membrane component (PGRMC) family. mPR, localized to the plasma membrane, rapidly activates intracellular signaling upon binding progesterone [20], functioning independently of nuclear PGR to regulate follicular development, ovulation, endometrial receptivity, and pregnancy maintenance. The PGRMC family, including PGRMC1 and PGRMC2, consists of membrane-associated proteins capable of binding progesterone and belongs to a distinct family from mPR [21]. These receptors also participate in rapid, non-genomic progesterone signaling [21].

During decidualization, differentiation-associated genes such as *HOXA10 (homeobox A10)*, *SOX4 (SRY-related high-mobility-group box 4)*, *HAND2 (heart and neural crest derivatives expressed 2)*, and *FOXO1 (Forkhead box protein O1)* are expressed, enabling the endometrium to acquire “receptivity” [22,23,24] (Figure 1). In particular, FOXO1 acts as a central transcription factor regulating differentiation processes, including insulin-like growth factor binding protein-1 (IGFBP1) and prolactin (PRL) secretion, thereby determining the transition from proliferation to differentiation and maturation [22,23,24,25,26,27,28]. Moreover, FOXO1 induces the expression of p21 and p27, leading to permanent cell cycle arrest, suppressing apoptosis, and thus modulating cellular senescence [28,29,30].

In endometriosis, reduced expression of PGR and impaired non-genomic pathways (mPR and PGRMC) have been reported, forming the molecular basis of progesterone resistance [20,31]. Furthermore, local estrogen excess suppresses the expression of PGR and HOXA10, thereby inhibiting progesterone-dependent differentiation responses [32,33]. Such hormonal imbalance results in decreased expression and functional impairment of decidualization-related transcription factors, including FOXO1, HOXA10, and HAND2, ultimately leading to defective decidualization. At the molecular level, impaired differentiation has been linked to inflammatory cytokines (tumor necrosis factor-α [TNF-α], interleukin-1 [IL-1]) [34], abnormalities in the WNT/β-catenin pathway and its inhibitor Dickkopf-related protein 1 (DKK1) [27], and phosphorylation-dependent destabilization of FOXO1 mediated by NEK2 (never in mitosis gene A-related kinase 2) [35]. Additionally, dysregulation of nuclear factor kappa-light-chain-enhancer of activated B cells (NF-κB) [36,37] and the phosphatidylinositol 3-kinase/protein kinase B (PI3K/AKT) signaling pathway [38,39] contributes to the maintenance of an undifferentiated cellular state. Thus, progesterone resistance in endometriosis arises from a multifactorial interplay of reduced hormone receptor expression, impaired non-genomic signaling, and chronic inflammation. Consequently, endometrial stromal cells fail to adequately activate the gene expression program required for decidualization, providing a major pathogenic basis for implantation failure and infertility.

### 2.2. Oxidative Stress and Inflammation in Decidualization

The pathophysiology of endometriosis is characterized by the interplay between chronic inflammation and oxidative stress. Local overexpression of aromatase and ERβ contributes to the establishment and maintenance of an inflammatory microenvironment, where persistent inflammatory responses are linked to pain, infertility, tissue fibrosis, and the acquisition of progesterone resistance [40]. Normally, endometrial tissues that reach the peritoneal cavity via retrograde menstruation are eliminated by macrophages and natural killer (NK) cells; however, in patients with endometriosis, the function of these immune cells is impaired, promoting the implantation and persistence of ectopic endometrial tissue [40,41,42]. As a result, inflammatory cytokines such as interleukin-1β (IL-1β), IL-6, and tumor necrosis factor-α (TNF-α) are continuously produced, activating the NF-κB signaling pathway and inducing overexpression of cyclooxygenase-2 (COX-2) and prostaglandin E_2_ (PGE_2_) [43]. This process amplifies inflammation, angiogenesis, and pain perception, ultimately forming an inflammatory feedback loop [44,45,46].

Moreover, endometriosis is characterized by a type 2 inflammatory response, with predominant roles of M2 macrophages, regulatory T (Treg) cells, and T helper type 2 (Th2) cells, along with immune suppression and tissue remodeling mediated by transforming growth factor-β (TGF-β) and IL-10 [47]. Endometriotic lesions exhibit an accumulation of M2-polarized macrophages [47]. M2 macrophages, also referred to as alternatively activated macrophages, are induced by IL-4, IL-10, IL-13, and TGF-β, and contribute to anti-inflammatory responses, tissue repair, angiogenesis, and fibrosis [48]. In addition, Treg cells are increased in the peritoneal cavity of endometriosis patients, promoting M2 macrophage polarization and creating an immune-tolerant environment via TGF-β, thereby facilitating immune evasion of ectopic endometrium [49].

Oxidative stress is closely linked to chronic inflammation and represents another major driver of disease progression. Reactive oxygen species (ROS) increase as a result of ovulation, retrograde menstruation, hemoglobin and iron in menstrual blood, inflammation, obesity, and mitochondrial dysfunction; when the antioxidant capacity is exceeded, oxidative stress ensues [50,51,52] (Figure 2). ROS induce single- and double-stranded DNA breaks, triggering a DNA damage response (DDR) centered on the ataxia-telangiectasia mutated (ATM)/ATM- and Rad3-related (ATR)-p53 pathway, which leads to the generation of the histone modification marker γ-H2AX, cell cycle arrest, apoptosis, and cellular senescence [53,54,55,56,57]. ROS stabilize p53, inducing the cyclin-dependent kinase inhibitor p21, which promotes G1/S cell cycle arrest and reinforces senescence programs [58]. Furthermore, in endometrial stromal cells, ROS generated under hypoxic conditions activate p38 mitogen-activated protein kinase (p38MAPK) and NF-κB, sustaining inflammatory gene expression and linking the damage response with inflammation [57,59]. Thus, oxidative stress and inflammation mutually amplify one another, contributing to disease persistence and progression.

Oxidative stress also affects endometrial differentiation and decidualization. ROS suppress PGR transcriptional activity via small ubiquitin-like modifier (SUMO) modification, thereby inhibiting decidualization [60]. Conversely, FOXO1 induces transcription of antioxidant enzymes such as superoxide dismutase 2 (SOD2), conferring resistance to ROS-induced cell death [61]. Additionally, ROS stimulate IL-33 secretion from ectopic endometrial stromal cells, promoting β-catenin activation and expression of epithelial–mesenchymal transition (EMT)-related transcription factors, including Snail, Twist, and Zeb1 (zinc finger E-box binding homeobox 1). Consequently, cellular migration and invasion capacities are enhanced, driving endometrial cells toward a more aggressive phenotype [62,63,64,65]. The synergistic effect of oxidative stress and estrogen signaling is also evident, as ROS activate survival and proliferation pathways such as extracellular signal-regulated kinase (ERK) [66] and PI3K/protein kinase B (AKT)/mammalian target of rapamycin (mTOR) [67]. These effects suppress apoptosis and facilitate the survival and expansion of ectopic endometrial tissue. Therefore, the sustained oxidative stress promotes ectopic cell survival through the induction of senescence [55], promotion of EMT [62,63,64,65], and inhibition of apoptosis [66,67]. Indeed, elevated levels of oxidative stress markers such as ROS, malondialdehyde (MDA), and 8-hydroxy-2′-deoxyguanosine (8-OHdG) have been consistently reported in the peritoneal fluid of patients [50,51,65,68,69], indicating a microenvironment prone to DNA oxidative damage. This suggests that oxidative stress is not merely a consequence but also a pathogenic factor in endometriosis.

In summary, chronic inflammation and oxidative stress act synergistically to drive disease progression by disrupting immune responses, altering hormonal signaling, promoting fibrosis and EMT, and impairing decidualization through cellular senescence. Thus, these processes should be regarded as central mechanisms that simultaneously represent both the “cause” and the “consequence” of endometriosis.

### 2.3. Metabolic Reprogramming

Endometrial cells or endometriotic cells that reflux into the peritoneal cavity require adenosine triphosphate (ATP) to adapt to the harsh microenvironment characterized by chronic inflammation and hypoxia [70,71]. Under hypoxic conditions, activation of glycolysis becomes essential, during which hypoxia-inducible factor-1α (HIF-1α) is stabilized and translocates into the nucleus to activate the transcription of glycolytic genes such as *glucose transporter 1 (GLUT1)*, *hexokinase 2 (HK2)*, and *lactate dehydrogenase A (LDHA)* [70,71] (Figure 3). In endometriosis, the expression of genes associated with the mitochondrial electron transport chain is reduced, whereas glycolytic enzymes such as HK2, pyruvate kinase, and 6-phosphofructo-2-kinase/fructose-2,6-bisphosphatase 4 (PFKFB4) are upregulated [72,73,74]. Similarly, studies in non-human primate models have demonstrated impaired oxidative phosphorylation and a metabolic shift toward glycolysis [75]. These findings indicate that endometriotic cells preferentially utilize anaerobic glycolysis, enabling the rapid generation of ATP under hypoxic conditions despite its lower efficiency. Furthermore, similar to tumor cells, they undergo “metabolic reprogramming” and exhibit the Warburg effect [73]. Excess estrogen within lesions further promotes metabolism by activating the PI3K/AKT/mTOR pathway, which enhances glycolytic enzyme expression and stabilizes HIF-1α [73]. The inflammatory milieu also imposes sustained energy demands; cytokines such as IL-6 and TNF-α activate the NF-κB and signal transducer and activator of transcription 3 (STAT3) pathways [76], thereby promoting fibrosis [77] and reinforcing glycolysis through the induction of GLUT1 and LDHA [78,79]. NF-κB directly promotes the transcription of glycolytic genes such as *GLUT1*, *HK2*, and *PFKFB3*, regulating the balance between glycolysis and mitochondrial respiration [80,81].

In addition, metabolic intermediates are linked to epigenetic regulation. Elevated lactate production resulting from enhanced glycolysis induces histone lactylation, which promotes cell proliferation and invasion [82]. Moreover, the long non-coding RNA (lncRNA) H19 facilitates aerobic glycolysis and mediates histone modifications through lactate production [83]. Thus, DNA methylation and histone modifications regulate the transcription of glycolysis-related genes, bridging metabolism and gene expression.

In summary, enhanced glycolytic activity in endometriotic cells represents a compensatory mechanism that ensures rapid ATP supply under hypoxia. This metabolic adaptation is supported by a complex interplay of metabolic reprogramming, inflammatory signaling, and epigenetic modifications.

### 2.4. Disruption of Cellular Homeostasis in Endometriosis

In endometriosis, multiple cellular homeostatic mechanisms, including cellular senescence, apoptosis, and autophagy, are collectively impaired, thereby contributing to lesion formation and progression.

#### 2.4.1. Enhanced Cellular Senescence and Deficiency of Clearance Mechanisms

Cellular senescence is a controlled state of cell cycle arrest that is normally induced by DNA damage or telomere shortening through the p53–p21 or p16–Rb pathways, contributing to tissue homeostasis and tumor suppression [84,85] (Figure 4). In the normal endometrium, some cells undergo transient senescence during decidualization and thereby participate in tissue remodeling, but senescent cells are usually eliminated by apoptosis [85]. In endometriotic lesions, however, p53 frequently becomes dysfunctional due to Mouse Double Minute 2 homolog (MDM2) overexpression, and the expression of p16/p21 is unstable, leading to the accumulation of aberrant cells that continue to survive [85,86,87]. In other words, excessive activation of MDM2 suppresses the function of p53 and contributes to disease progression by inhibiting apoptosis and promoting the persistent survival of ectopic cells. Meanwhile, chronic DNA damage induced by ROS and iron metabolism abnormalities persists in endometriotic lesions, thereby activating the p53–p21 pathway [58,88,89] and promoting the acquisition of a senescent phenotype characterized by the expression of markers such as p16 and senescence-associated β-galactosidase (SA-β-gal) [69]. Senescent decidual cells secrete a senescence-associated secretory phenotype (SASP), which consists of diverse factors including interleukins (IL-6, IL-8, IL-1β, TNF-α, IL-10, IL-12, IL-11), chemokines (C-C motif chemokine ligand 2 (CCL2), CCL24), growth factors (Vascular endothelial growth factor (VEGF), fibroblast growth factor 2 (FGF2), growth differentiation factor 15 (GDF15), stanniocalcin-1 (STC1)), proteases (matrix metalloproteinase (MMP)-1, MMP-3), and serine proteinase inhibitors (SERPINs) [21,87,90,91,92]. SASP modifies the function of surrounding cells and the immune response, sustaining an inflammatory microenvironment. Normally, SASP promotes the recruitment of uterine natural killer (uNK) cells, which eliminate senescent cells via phagocytosis. However, in endometriosis, both the number and function of uNK cells are diminished, resulting in persistent SASP, chronic inflammation, and impaired decidualization [93,94,95,96]. The local recruitment of uNK cells involves cytokines and chemokines such as IL-15 and C-X-C motif chemokine ligand 12 (CXCL12), and aberrant expression of these molecules in endometriosis may further impair uNK accumulation within the endometrium [96]. Through SASP, endometriotic cells propagate senescence and inflammation to neighboring cells, thereby amplifying the chronic inflammatory milieu and contributing to fibrosis, pain, and infertility [21,87,90,97,98,99].

#### 2.4.2. Evasion of Apoptosis and Enhancement of Survival Signals

Ectopic endometrial cells continue to survive and proliferate despite being normally destined for elimination. This persistence is supported by the upregulation of anti-apoptotic factors such as B-cell lymphoma 2 (BCL-2) and BCL-xL, along with the downregulation of pro-apoptotic mediators such as BCL2 associated X, apoptosis regulator (BAX), and caspase-3 [100]. Furthermore, constitutive activation of the PI3K/AKT pathway promotes cytoplasmic translocation of FOXO1 and downregulation of IGFBP1, thereby inhibiting decidualization [39]. Inflammatory cytokines (TNF-α, IL-6, IL-1β) activate NF-κB, which induces anti-apoptotic genes including *BCL2* and *X-linked inhibitor of apoptosis protein* (*XIAP*) [101,102]. Estrogen (via ERβ) synergistically enhances cell survival and proliferation, while progesterone resistance disrupts p53 signaling regulation [6,92]. Collectively, these mechanisms enable ectopic cells to evade apoptosis and maintain survival.

#### 2.4.3. Dysfunction of Autophagy

Autophagy is a cytoprotective mechanism that allows cells to evade apoptosis [103]. It is a lysosome-dependent degradative process that recycles unnecessary or damaged intracellular components in response to environmental stresses such as nutrient deprivation and oxidative stress, thereby maintaining cellular homeostasis, nutrient balance, stress responses, and survival [104]. In the normal endometrium, autophagy prevents the accumulation of toxic substances, regulates cyclic endometrial remodeling, supports decidualization, and participates in oocyte maturation, embryo implantation, and fetal development, suggesting its involvement in reproductive aging [104,105,106]. Conversely, impaired autophagy contributes to infertility and reproductive failure, while age-associated accumulation of abnormal proteins promotes reproductive aging [107].

Autophagy is tightly regulated through a dynamic network centered on p53, AMP-activated protein kinase (AMPK), and mTOR [108] (Figure 5). Regulatory components include the unc-51 like autophagy activating kinase 1 (ULK1) complex, Beclin-1 complex, transcription factor EB (TFEB), FOXO1/3, sirtuin 1 (SIRT1), HIF-1α, ROS, microRNAs, and epigenetic modifications [109,110,111]. In the normal endometrium, p53 binds to response elements in the PTEN promoter to suppress the PI3K/AKT pathway, thereby contributing to apoptosis induction and tumor suppression [112]. Conversely, activation of PI3K/AKT promotes p53 degradation via MDM2, thereby inhibiting apoptosis and cell cycle arrest [113]. Moreover, p53 activates BAX and P53 up-regulated modulator of apoptosis (PUMA), altering mitochondrial outer membrane permeability and initiating the caspase cascade. In the normal endometrium, p53 fluctuates according to the menstrual cycle, increasing particularly during menstruation to induce BAX and PUMA. In endometriosis, reduced p53 expression suppresses autophagy, facilitates apoptosis evasion, impairs decidualization, promotes DNA damage accumulation, and sustains an inflammatory microenvironment [114]. AMPK serves as an intracellular energy sensor that is activated under ATP depletion and AMP accumulation. Activated AMPK suppresses mTOR to induce autophagy and reinforces its self-activation by promoting liver kinase B1 (LKB1) phosphorylation through SIRT1 [115,116,117]. Under hypoxic conditions in the endometrium, the HIF-1/AMPK pathway is activated and contributes to autophagy induction during menstruation [118]. However, in endometriosis, excessive estrogen and hyperactivation of mTOR inhibit AMPK function, thereby suppressing autophagy and promoting the survival of ectopic cells [67]. mTOR complex 1 (mTORC1) is a central regulator of cell growth and proliferation and also inhibits autophagy. ROS stimulate the PI3K/AKT pathway through PTEN inactivation [119]. PI3K is activated by growth factors and insulin, which then phosphorylate and inactivate tuberous sclerosis complex 2 (TSC2) via AKT, thereby activating Ras homolog enriched in brain (Rheb) and stimulating mTORC1 [120,121,122,123,124]. REDD1 (regulated in development and DNA damage responses 1) activates the TSC1/TSC2 complex to inhibit Rheb, thus suppressing mTORC1. However, ROS-induced degradation or inhibition of REDD1 leads to increased mTOR activity [123]. While short-term ROS exposure may induce autophagy, chronic oxidative stress acts suppressively [67,125]. Therefore, autophagy is multilayeredly regulated by nutritional status, hormonal environment, oxidative stress, and inflammation. In endometriosis, hyperestrogenic conditions and chronic inflammation cause sustained mTOR activation, NF-κB positive feedback, and constitutive stimulation of the PI3K/AKT/mTOR and MAPK/ERK pathways, resulting in autophagy suppression, abnormal proliferation, lesion maintenance, and infertility [126,127,128,129,130,131,132,133].

##### Reduction in Autophagy

In endometriosis, reduced expression and activity of autophagy-related proteins have been reported compared with normal tissue [105]. Aberrant autophagic flux and dysregulated expression of multiple factors are observed in ectopic tissues, with peritoneal fluid analyses showing increased mRNA expression of PI3K, FLICE-inhibitory protein (FLIP), and Rubicon [134]. Hyperactivation of the PI3K/AKT/mTOR pathway promotes ectopic cell survival, FLIP suppresses apoptosis by inhibiting caspase-8, and Rubicon blocks autophagosome maturation [134]. Furthermore, reduced expression of LC3 (microtubule-associated protein 1 light chain 3) and Beclin-1 has been reported, with Beclin-1 levels showing a negative correlation with serum CA125 levels and pelvic pain severity, suggesting its contribution to disease progression [135,136,137,138]. Overall, endometriosis is characterized by a molecular environment dominated by autophagy suppression [67,103]. Moreover, BCL-2/adenovirus E1B 19 kDa-interacting protein 3 (BNIP3), induced by HIF-1, promotes mitochondrial mitophagy, thereby supporting cell survival and metabolic adaptation. However, in a rat endometriosis model, the expression of BNIP3, BECN1, and LC3II was found to be reduced, indicating impaired autophagy and mitophagy [138].

##### Enhancement of Autophagy

Conversely, some studies have reported enhanced autophagy. Increased Beclin-1 and LC3 expression, along with decreased p62 and p53 expression, have been observed in patient samples and rat models, suggesting a potential role of autophagy in suppressing apoptosis under environmental stress conditions such as hypoxia and iron overload [4,5,103,139,140]. Autophagy is regulated not only by the PI3K/AKT/mTOR pathway but also by external factors such as hypoxia and iron loading [103]. While hypoxia and iron stress can activate autophagy, mTOR activation may suppress both autophagy and apoptosis [103]. These processes are closely linked to the survival and persistence of ectopic cells [4]. Autophagic activity may fluctuate temporally during disease progression: being activated in early stages to support cell survival, but suppressed during chronic stages [141,142]. During suppression, the accumulation of damaged structures promotes inflammation, proliferation, and angiogenesis, contributing to lesion progression. Conversely, in ovarian endometriotic cysts with high oxidative stress, sustained autophagy activation may also promote cell survival. These contradictory findings likely reflect differences in disease stage, tissue microenvironment, and the complexity of signaling networks [103]. Reports examining both autophagy and ferroptosis suggest that in early lesions, both processes are transiently activated in response to hypoxia and iron overload. However, persistent stimulation by estrogen and the mTOR pathway subsequently suppresses them, thereby promoting lesion chronicity [142]. Furthermore, overexpression of SLC7A11 (solute carrier family 7 member 11) in endometriosis enables cells to escape iron-dependent ferroptosis, allowing ectopic cells to survive even under oxidative stress [143].

Autophagy also plays a role in the aging processes of reproductive tissues, including the ovary, oocytes, and endometrium, and is thought to contribute critically to so-called “reproductive aging” [21,87,90,91,92]. Reproductive aging refers to the age-related decline in ovarian reserve, characterized by a reduction in both the quantity and quality of oocytes, accompanied by hormonal changes and impaired endometrial function. Thus, a common molecular basis mediated by autophagy may underlie both endometriosis-associated reproductive dysfunction and age-related fertility decline. Dysregulated autophagy impairs apoptosis and decidualization, thereby contributing to infertility, while age-related decline in autophagy compromises oocyte cytoplasmic quality and embryonic developmental competence [6,144]. Mitophagy maintains oocyte cytoplasmic quality by eliminating damaged mitochondria and misfolded proteins [145], and studies in ATG7-deficient mice have demonstrated premature depletion of the primordial follicle pool [146]. Accordingly, autophagy activation extends follicle lifespan, delays the decline in ovarian reserve, and improves embryo development rates, whereas its decline leads to DNA damage accumulation, chronic inflammation, and reduced reproductive capacity [145,146,147]. Taken together, both reduced autophagy and excessive or imbalanced activation represent shared molecular mechanisms in endometriosis and age-related infertility, contributing to disease progression and reproductive aging [104,148,149].

In summary, these four processes—enhanced cellular senescence, sustained SASP activity, apoptosis evasion, and autophagy dysfunction—act in concert as major mechanisms that comprehensively support the survival and invasion of lesion cells in endometriosis, as well as the development of infertility, impaired decidualization, pain, and fibrosis.

### 2.5. Genetic Regulation in Decidualization

Epigenetic regulatory mechanisms include DNA methylation, histone modifications, and non-coding RNAs such as microRNAs (miRNAs) and lncRNAs. In endometriosis, impaired decidualization is observed not only in ectopic lesions but also in eutopic endometrium, involving progesterone and PGE_2_ signaling defects mediated by hormone resistance, chronic inflammation, and epigenetic alterations [21,93]. In particular, hypermethylation of the progesterone receptor-B (PGR-B) promoter and abnormalities in DNA methylation and histone modifications lead to decreased PGR expression [32,150,151]. Conversely, hypomethylation of the ERβ promoter results in ERβ overexpression, which disrupts the balance with ERα and further enhances PGR suppression [33]. Downstream progesterone-responsive genes, such as *HOXA10*, *IGFBP1*, and *FOXO1*, are also affected by DNA methylation and histone modifications, and are altered in endometriosis [152].

HOXA10 expression is normally upregulated by progesterone in the endometrium to promote receptivity, but in endometriosis, its expression is restricted by promoter methylation, potentially contributing to implantation failure [153]. IGFBP1, a marker of decidualization induced by progesterone and cyclic adenosine monophosphate (cAMP), is downregulated in endometriosis due to overexpression of enhancer of zeste homolog 2 (EZH2) [154]. FOXO1 is normally induced by estradiol (E2), medroxyprogesterone acetate (MPA), and cAMP, but in endometriosis it is suppressed via the SIRT1 and p53 pathways, contributing to defective decidualization and persistent inflammation [155,156]. Furthermore, N6-methyladenosine (m^6^A) modification by methyltransferase-like 3 (METTL3) promotes the degradation of FOXO1 mRNA, thereby impairing decidualization and embryo implantation [157]. Reduced expression of Notch signaling-related genes also contributes to FOXO1 downregulation [158].

DNA methyltransferases (DNMT1, DNMT3A, and DNMT3B) fluctuate in expression across the menstrual cycle and regulate genes associated with decidualization. Estrogen has been reported to increase DNMT activity and thereby contribute to PGR suppression [159,160]. In ectopic endometrium, both overexpression and underexpression of DNMTs have been reported, suggesting individual or tissue-specific differences [160,161].

MicroRNAs also play a role in post-transcriptional regulation. miR-194 contributes to progesterone resistance and reduced fertility by suppressing decidualization [162]. miR-375 inhibits NADPH oxidase 4 (NOX4) expression and reduces ROS production, thereby impairing decidualization [163]. miR-29c suppresses FK506-binding protein 4 (FKBP4), affecting the maturation of the progesterone receptor complex [164]. miR-196a activates the mitogen-activated protein kinase/extracellular signal-regulated kinase (MEK/ERK) pathway, suppresses progesterone expression, and contributes to decreased progesterone sensitivity in endometriosis [165].

In summary, excessive estrogen, inflammatory cytokines, aberrant DNA methylation, histone modifications, and altered miRNA expression collectively contribute to suppression of PGR—particularly PGR-B—in endometriosis, leading to imbalanced progesterone and estrogen signaling [64,166]. ERβ overexpression enhances NF-κB activation, which promotes inflammation while further strengthening PGR suppression. These molecular alterations form the basis of reduced progesterone responsiveness, representing the underlying mechanism of progesterone resistance in endometriosis [166].

## 3. Decidualization as a Selection Barrier for Embryo Quality

Decidualization and cellular senescence function as preparatory mechanisms to ensure successful pregnancy. Moreover, senescence in a subset of decidualized cells may act as a selection mechanism for assessing embryo quality [12,84] (Figure 6). Inflammatory factors secreted by senescent cells, known as the SASP, may be tolerated by high-quality embryos, whereas they can act as excessive stress signals that hinder the implantation of low-quality embryos. This suggests that the endometrium incorporates cellular senescence as a stress-induced environment to selectively accept competent embryos. Thus, spontaneous decidualization not only prepares the endometrium for implantation but also functions as a maternal selection system that rejects poor-quality embryos, effectively serving as a form of quality control in pregnancy [167].

In the endometrium, persistent senescent cells may exert pathogenic effects, contributing to inflammation and tumorigenesis. However, these cells are transient and localized, subsequently being cleared primarily by immune cells such as uNK cells, thereby enabling tissue regeneration [11,168]. This mechanism is considered a physiologically rational system, as it endows reproductive organs with self-repair capacity.

In contrast, endometriosis is associated with various abnormalities in endometrial function, including impaired decidualization and excessive cellular senescence. These alterations disrupt embryo implantation, compromise pregnancy sustainability, and ultimately reduce fertility. Even when pregnancy is achieved, defective placentation and impaired uterine environment increase the risk of obstetric complications, including miscarriage, preterm birth, fetal growth restriction, and preeclampsia [169]. Therefore, endometriosis should not be regarded merely as a pain-associated disorder but as a disease intricately linked to pregnancy and childbirth. At its core lies the disruption of finely regulated biological processes such as decidualization and cellular senescence.

## 4. Discussion and Future Perspectives

This review highlights the complex pathophysiology and molecular mechanisms of endometriosis, with particular focus on dysregulation of the p53–AMPK–mTOR pathway and its interplay with oxidative stress, inflammation, metabolic reprogramming, apoptosis, and autophagy, alongside abnormalities in decidualization and cellular senescence. Retrograde endometrial cells are exposed to hostile conditions such as hypoxia and inflammation, yet those able to evade apoptosis survive and contribute to lesion formation. In the early stages, autophagic activity is transiently elevated [4,104] as an adaptive stress response, but this response often diminishes with chronicity, contributing to impaired decidualization and infertility. Reduced activity of p53 and AMPK, coupled with hyperactivated mTOR signaling, synergizes with elevated estrogen levels, oxidative stress, chronic inflammation, and epigenetic alterations to suppress autophagy and accelerate senescence, thereby promoting lesion progression.

Clinically, endometriosis is strongly associated with infertility, implantation failure, miscarriage, and obstetric complications [169]. Reduced endometrial receptivity is linked to progesterone resistance, inflammation, oxidative stress, aberrant DNA methylation and histone modifications, and impaired embryo–endometrium crosstalk [170,171,172]. Inappropriate cellular senescence resulting from diminished autophagy is increasingly recognized as a key contributor to fertility decline. Evidence suggests that pharmacological interventions such as rapamycin may reduce oxidative stress and senescence markers while improving IVF outcomes, although adverse effects must be considered [173,174]. Agents that promote autophagy may also improve oocyte quality [175]. Thus, targeting endometrial factors such as defective decidualization and aberrant senescence holds promise for reducing reliance on assisted reproductive technology (ART) and enhancing fertility. Moreover, abnormalities in autophagy have also been reported in polycystic ovary syndrome (PCOS), premature ovarian insufficiency (POI), and pregnancy complications such as hypertensive disorders and gestational diabetes mellitus [176,177,178], indicating that ovarian function and endometrial senescence outside of pregnancy may modulate the severity of these conditions.

During decidualization, a subset of stromal cells undergoes transient senescence and contributes to tissue remodeling via secretion of the senescence-associated secretory phenotype (SASP). While this process is essential for normal decidual formation, excessive or prolonged senescence disrupts this balance, leading to pathological outcomes. Autophagy plays a pivotal role in maintaining this equilibrium and is indispensable for reproductive function. Disruption of the coordinated coexistence of transient senescence and tissue remodeling, a process unique to human reproduction, contributes directly to endometriosis and infertility.

Future research should systematically elucidate the temporal and spatial dynamics of the p53–AMPK–mTOR axis, autophagy, and senescence from the onset of endometriosis through lesion progression to endometrial functional decline, thereby clarifying causal relationships with infertility. Comprehensive analyses of epigenetic changes in both ectopic and eutopic endometrium, including those involving miRNAs and DNA methylation patterns, are expected to identify novel diagnostic biomarkers and prognostic indicators. From a therapeutic perspective, promising strategies include pharmacological suppression of hyperactivated signaling pathways such as mTOR and NF-κB, activation of AMPK or p53, and interventions that induce autophagy. The development of biomarkers capable of assessing decidualization status and senescence activity may further enable personalized approaches for infertility management.

Endometriosis should therefore be regarded not only as a disorder of pelvic pain but as a systemic disease that profoundly impacts female reproductive health. Advances in basic and translational research focusing on autophagy and senescence are essential for developing innovative therapies. Ultimately, the establishment of therapeutic strategies aimed at restoring autophagy regulation and suppressing aberrant senescence may improve fertility outcomes without excessive reliance on ART, representing a transformative step in clinical management and reproductive medicine.

## 5. Methods

### Search Strategy and Selection Criteria

This study encompasses a wide range of approaches, including animal models, in vitro experiments, and various clinical studies, resulting in heterogeneity in the content and quality of the literature. In light of the diversity and complexity of the research perspectives, we adopted a narrative review format rather than a systematic review. To ensure the relevance of the selected studies, we applied explicit inclusion and exclusion criteria and included only those articles deemed highly pertinent. Our search strategy employed keywords such as “p53”, “AMPK”, “mTOR”, “endometriosis”, “cellular senescence”, and “decidualization”, which were combined using Boolean operators to enhance search precision, as outlined in Table 1. Literature searches were conducted using PubMed (https://pubmed.ncbi.nlm.nih.gov/) and Google Scholar (https://scholar.google.com/) with appropriate combinations of text words. The search period extended from the inception of each database through April 2025. We included original research articles and review papers published in English, as well as references cited within those articles. Publications in languages other than English were excluded. During the initial screening phase, duplicate records identified through database and manual searches were removed. Subsequently, study eligibility was assessed based on titles and abstracts. Full-text evaluations were independently performed by five authors (S.I., M.N., M.U., H.S., and H.H.). In cases of disagreement or ambiguity regarding study selection, consensus was reached through discussion among the authors. In the final selection phase, studies lacking sufficient basic or clinical data, or those with unclear relevance to the pathophysiology of endometriosis, were excluded. However, studies focused on cancers or other diseases were included if they provided strong implications for understanding the pathophysiology of endometriosis, even if not directly related. The overall selection process and detailed inclusion and exclusion criteria are summarized in the flowchart presented in Figure 7.

## 6. Conclusions

Endometriosis is driven by complex molecular interactions in which dysregulated p53–AMPK–mTOR signaling, impaired autophagy, and aberrant cellular senescence converge to disrupt decidualization and compromise reproductive function. These processes are further amplified by oxidative stress, inflammation, hormonal imbalance, and epigenetic alterations, underscoring the systemic nature of the disease. Future strategies that restore autophagy and normalize senescence, while targeting key signaling and epigenetic pathways, hold promise for improving fertility outcomes and reducing reliance on assisted reproductive technologies.

## Figures and Tables

**Figure 1 ijms-26-09125-f001:**
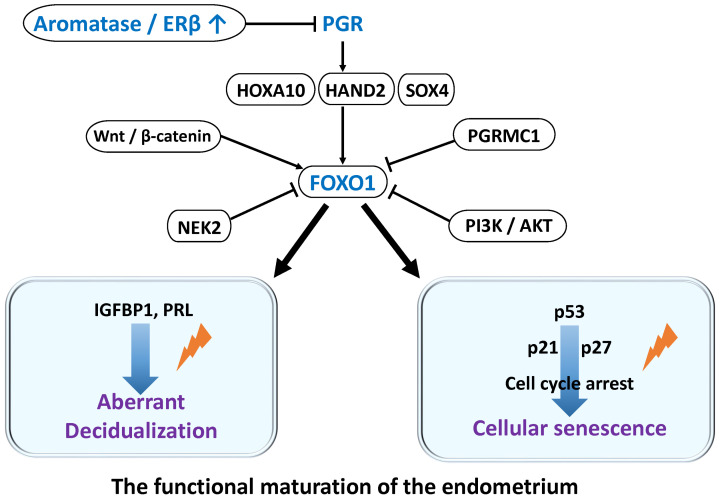
FOXO1-Centered Decidualization and Differentiation Defects in Endometrial Stromal Cells: Association with Endometriosis. Decidualization of endometrial stromal cells (ESCs) critically depends on progesterone-induced FOXO1 expression and activation of differentiation-related genes. FOXO1 regulates not only differentiation but also cell cycle arrest and cellular senescence, controlling heterogeneity in decidualization. In endometriosis, inflammatory cytokines, estrogen excess, signaling abnormalities, and NEK2 activation reduce FOXO1 expression and stability, causing differentiation defects and impaired decidualization. These contribute to implantation failure and infertility. The lightning symbol denotes impaired processes of decidualization and cellular senescence.

**Figure 2 ijms-26-09125-f002:**
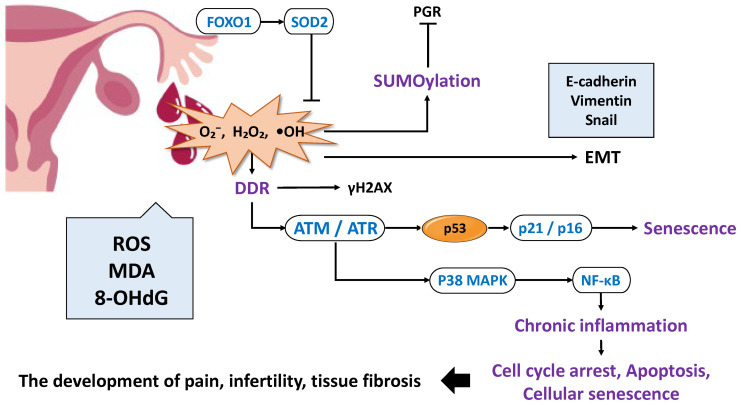
Pathogenic Role of Oxidative Stress in Endometriosis. This figure focuses on the impact of oxidative stress on endometrial cells and discusses its role in the onset and progression of endometriosis. Excessive ROS production induces DNA damage response, p53 activation, cell cycle arrest, and senescence, while promoting inflammation and angiogenesis via NF-κB and MAPK pathways. Oxidative stress also affects decidualization and hormone responsiveness through SUMOylation and FOXO1 regulation. Furthermore, ROS induce epithelial–mesenchymal transition and epigenetic alterations, facilitating the transition of endometrial cells to an endometriosis-like phenotype. These changes are closely linked to lesion formation and maintenance, suggesting oxidative stress plays a causative role in endometriosis.

**Figure 3 ijms-26-09125-f003:**
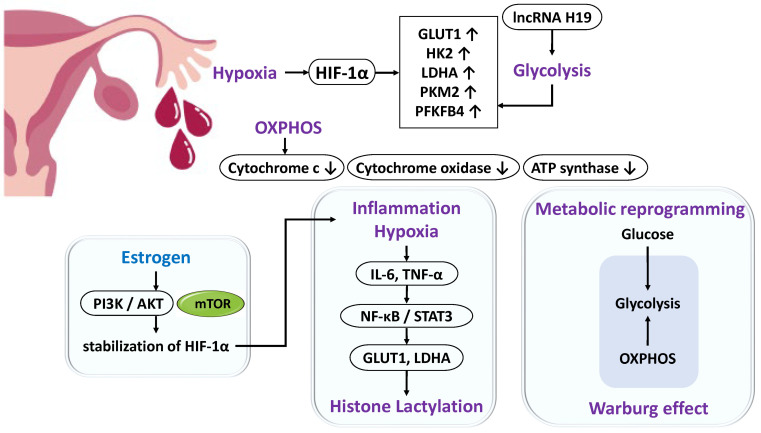
Metabolic Reprogramming in Endometriosis: Enhanced Glycolysis and Its Molecular Mechanisms. Endometriotic cells adapt to the hypoxic and chronic inflammatory environment of the peritoneal cavity by metabolic reprogramming that enhances glycolysis, an oxygen-independent pathway. HIF-1α and estrogen-dependent PI3K/AKT/mTOR signaling promote expression of glycolysis-related genes such as *GLUT1* and *HK2*. Activation of NF-κB and STAT3 by IL-6 and TNF-α also contributes to this process. Additionally, epigenetic regulation, including histone lactylation induced by lactate, increases cellular activity and facilitates lesion progression.

**Figure 4 ijms-26-09125-f004:**
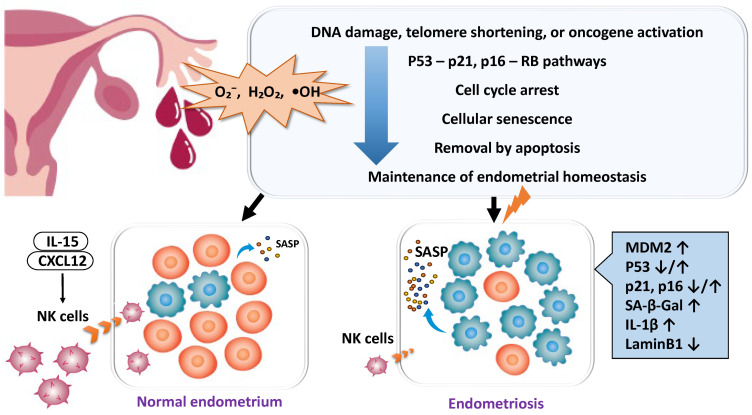
Role of Cellular Senescence in Endometriosis and Its Impact on Pathogenesis. Cellular senescence is an important homeostatic mechanism contributing to tumor suppression, but its regulation is disrupted in endometriosis. While DNA damage and oxidative stress induce senescence in lesions, p53 pathway dysfunction and unstable senescence marker expression lead to the accumulation of senescent cells. These cells secrete inflammatory factors that promote chronic inflammation, fibrosis, decidualization defects, and infertility. Although senescence normally functions physiologically, its pathological persistence in endometriosis may cause implantation failure and miscarriage. The decidual cells and senescent cells depicted in this figure are similar to those described in reference [6], but they are not identical. The lightning symbol denotes impaired processes of decidualization and cellular senescence.

**Figure 5 ijms-26-09125-f005:**
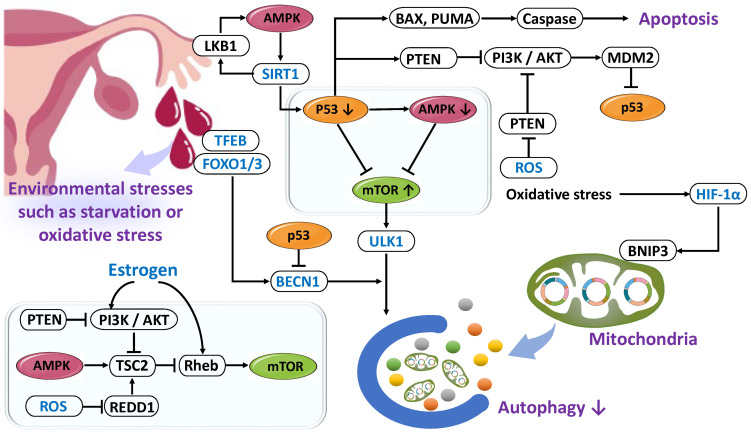
Regulation of Autophagy and the Roles of p53, AMPK, and mTOR Signaling in Endometriosis. Autophagy is regulated by a complex molecular network centered on p53, AMPK, and mTOR, playing essential roles in cellular homeostasis and senescence prevention. In endometriosis, decreased p53 expression, suppressed AMPK function, and excessive mTOR activation inhibit autophagy, leading to apoptosis resistance, persistent inflammation, and decidualization defects. This pathophysiology promotes ectopic endometrial cell survival and proliferation, contributing to infertility and lesion progression. mTOR inhibition is attracting attention as a therapeutic target. The symbols for p53, AMPK, and mTOR used in this figure are the same as those in reference [6].

**Figure 6 ijms-26-09125-f006:**
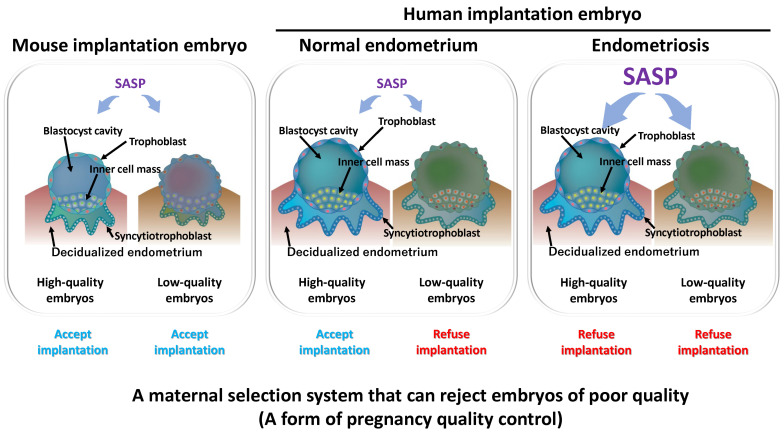
Evolutionary Significance of Human-Specific Natural Decidualization and Cellular Senescence and Their Relationship to Endometriosis. This figure illustrates the process of implantation from left to right: implantation embryo in mice, in normal human endometrium, and in human endometrium affected by endometriosis. In each panel, the left side represents the implantation pattern of a high-quality (viable) embryo, while the right side depicts that of a low-quality (non-viable) embryo. (**Left**): Placentation in mice: In mice, the uterine environment is relatively permissive, allowing even low-quality embryos to implant and proceed with placental development without significant hindrance. (**Middle**): Placentation in normal human endometrium: In a physiologically normal human endometrium, implantation and placental development proceed effectively when the embryo is of high quality. However, implantation of low-quality embryos is generally restricted. (**Right**): Placentation in human endometrium with endometriosis: In endometriosis-affected endometrium, decidualization is often impaired, which compromises the endometrium’s receptivity. As a result, even high-quality embryos may fail to implant properly. This suggests that impaired decidualization in endometriosis may reduce the likelihood of implantation not only for low-quality embryos but also for those of good quality.

**Figure 7 ijms-26-09125-f007:**
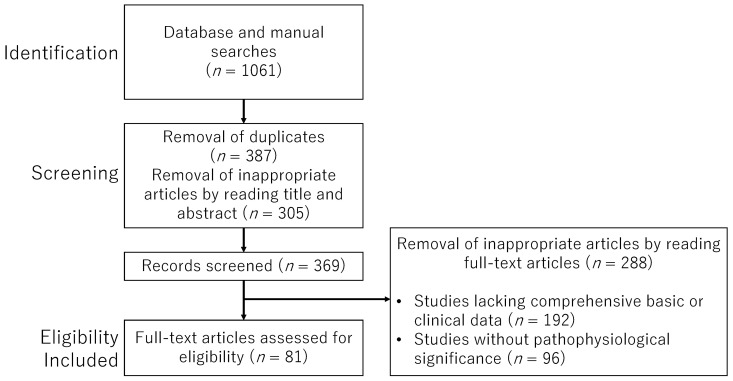
The flowchart outlines the study selection process.

**Table 1 ijms-26-09125-t001:** The keyword and search term combinations.

Search Mode	The Keyword and Search Term Combinations
Search term 1	p53
Search term 2	AMPK
Search term 3	mTOR OR mTORC1 OR mTORC2
Search term 4	Endometriosis OR Ectopic endometrium
Search term 5	Cellular senescence
Search term 6	Decidualization
Search	(Search term 1 OR 2 OR 3) AND Search term 4
	(Search term 1 OR 2 OR 3) AND Search term 5
	(Search term 1 OR 2 OR 3) AND Search term 6
	(Search term 1 OR 2 OR 3) AND Search term 4 AND (Search term 5 OR Search term 6)
	Search term 4 AND Search term 5
	Search term 4 AND Search term 6

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
