# Peer review of "Balancing Decidualization, Autophagy, and Cellular Senescence for Reproductive Success in Endometriosis Biology"

_ijms, 2025, doi:10.3390/ijms26189125_

Round 1

Reviewer 1 Report

Comments and Suggestions for Authors

This is an extensive review study unravelled molecular mechanisms implicated to the pathogenesis of endometriosis. The authors have stated that this review focused mainly to the disruption of the p53-AMPK-mTOR signaling axis. However, along the manuscript the reader can find various pathogenetic mechanisms that contribute to the development and the maintenance of endometriosis. 

I don't have any queries for the authors. English is fine, the text runs smoothly and is easily understandable for a specialized in the field reader, references are appropriate, and the structure of this review study is the proper one. This review study is already very extensive for proposing additional chapter about possible pharmaceutical treatment aiming to these molecular mechanisms. 

Author Response

Editor-in-Chief

International Journal of Molecular Sciences (IJMS)

We would like to express our sincere gratitude to you and to the reviewers for providing insightful comments and constructive suggestions on our manuscript (Manuscript ID: ijms-3807200). We carefully considered each point raised during the review process and made substantial revisions to address the concerns. In revising the manuscript, we have endeavored to remove unnecessary repetition, condense the text where appropriate, and ensure that only the most scientifically significant aspects are emphasized.

All revisions have been marked using the track changes function in Microsoft Word, with the modified text highlighted in blue for clarity. A detailed point-by-point response to the reviewers’ comments is provided below. We believe that these revisions have considerably improved the clarity, scientific rigor, and overall quality of the manuscript.

We are truly grateful for the opportunity to resubmit our work after revision, and we sincerely hope that the revised version will now be suitable for publication in IJMS. We once again thank you for your time and consideration, and we look forward to your response.

Yours sincerely,

Hiroshi Kobayashi

E-mail: hirokoba@naramed-u.ac.jp

Point-by-point responses to reviewer comments

Reviewer 1

Comment 1:

This is an extensive review study unravelled molecular mechanisms implicated to the pathogenesis of endometriosis. The authors have stated that this review focused mainly to the disruption of the p53-AMPK-mTOR signaling axis. However, along the manuscript the reader can find various pathogenetic mechanisms that contribute to the development and the maintenance of endometriosis.

I don't have any queries for the authors. English is fine, the text runs smoothly and is easily understandable for a specialized in the field reader, references are appropriate, and the structure of this review study is the proper one. This review study is already very extensive for proposing additional chapter about possible pharmaceutical treatment aiming to these molecular mechanisms.

Response 1:

We sincerely thank you for taking the time to review our manuscript.

Reviewer 2 Report

Comments and Suggestions for Authors

In a majority of subsections there are a lot of repetitive informations, the generalized discussion that starts with the role of oxidative stress in cell cycle arrest, continuing to cellular senescence and apoptosis in decidualization, regulated by factors such as FOXO1 and/or NFKB as a key molecules determining the direction between decidualization and cellular senescence. Also, each section includes an interplay of variety of inflammatory cytokines in this process. The entire paper should be restructured to provide more clarity and with improve the flow of the text, with following subsections:

Hormonal regulation of decidualization (merge current subchapter 2.1. Progesterone Resistance, 2.6. Decidualization Deficiency)

Oxdative stress and inflammation in decidualization (merge current subchapters 2.2. Chronic Inflammation; 2.3. Oxidative Stress)

Metabolic reprogramming (2.5. Energy Metabolism)

Apoptosis/cellular senescence balance in decidualization (merge  current sections Promotion of Cellular Senescence; 2.8. Promotion of SASP; 2.9. Evasion of Apoptosis; 2.10. Autophagy Dysfunction)

Gene regulation in decidualziation (includes current sections 2.4. Epigenetic Modifications)

Also, even individual subsections are not clearly explained. One example: The entire section on oxidative stress is unclear, with information presented in a scattered manner and lacking a coherent narrative. It should be reformulated with a clearer description of key parts: effects on DNA damage and cell cycle inhibition; effect on EMT transition; epigenetic modifications: DNA methylation and histone modifications. Section 361-375 should be separate subsection called microRNA expression, not a part of epigenetic modifications section

Section 3. Reproductive Strategy and Autophagy repeats information from the beginning of the paper, about the general mechanisms involved in decidualization process (lines from 962 to 993).Instead the section should be named “Decidua as a selection boundary for embryo quality”

In general paper is too long, with a little coherence and should be extensively rewritten.

Author Response

Editor-in-Chief

International Journal of Molecular Sciences (IJMS)

We would like to express our sincere gratitude to you and to the reviewers for providing insightful comments and constructive suggestions on our manuscript (Manuscript ID: ijms-3807200). We carefully considered each point raised during the review process and made substantial revisions to address the concerns. In revising the manuscript, we have endeavored to remove unnecessary repetition, condense the text where appropriate, and ensure that only the most scientifically significant aspects are emphasized.

All revisions have been marked using the track changes function in Microsoft Word, with the modified text highlighted in blue for clarity. A detailed point-by-point response to the reviewers’ comments is provided below. We believe that these revisions have considerably improved the clarity, scientific rigor, and overall quality of the manuscript.

We are truly grateful for the opportunity to resubmit our work after revision, and we sincerely hope that the revised version will now be suitable for publication in IJMS. We once again thank you for your time and consideration, and we look forward to your response.

Yours sincerely,

Hiroshi Kobayashi

E-mail: hirokoba@naramed-u.ac.jp

Point-by-point responses to reviewer comments

Reviewer 2

Comment 1:

In a majority of subsections there are a lot of repetitive informations, the generalized discussion that starts with the role of oxidative stress in cell cycle arrest, continuing to cellular senescence and apoptosis in decidualization, regulated by factors such as FOXO1 and/or NFKB as a key molecules determining the direction between decidualization and cellular senescence. Also, each section includes an interplay of variety of inflammatory cytokines in this process. The entire paper should be restructured to provide more clarity and with improve the flow of the text, with following subsections:

Hormonal regulation of decidualization (merge current subchapter 2.1. Progesterone Resistance, 2.6. Decidualization Deficiency)

Oxdative stress and inflammation in decidualization (merge current subchapters 2.2. Chronic Inflammation; 2.3. Oxidative Stress)

Metabolic reprogramming (2.5. Energy Metabolism)

Apoptosis/cellular senescence balance in decidualization (merge  current sections Promotion of Cellular Senescence; 2.8. Promotion of SASP; 2.9. Evasion of Apoptosis; 2.10. Autophagy Dysfunction)

Gene regulation in decidualziation (includes current sections 2.4. Epigenetic Modifications)

Also, even individual subsections are not clearly explained. One example: The entire section on oxidative stress is unclear, with information presented in a scattered manner and lacking a coherent narrative. It should be reformulated with a clearer description of key parts: effects on DNA damage and cell cycle inhibition; effect on EMT transition; epigenetic modifications: DNA methylation and histone modifications. Section 361-375 should be separate subsection called microRNA expression, not a part of epigenetic modifications section

Response 1:

We greatly appreciate your valuable comments. In the revised version, we reorganized Section 2 into five subsections (“2.1.”–“2.5.”), eliminated redundancies, shortened the text wherever possible, and refined the expressions for greater clarity. In Section 2.2, we added a description of the impact on DNA damage and cell cycle arrest in the third paragraph, and included an explanation of EMT-related transcription factors in the fourth paragraph. In Section 2.5, because the epigenetic regulatory mechanisms encompass DNA methylation, histone modifications, and non-coding RNAs, we did not create a separate subsection solely for “microRNA expression.”

Comment 2:

Section 3. Reproductive Strategy and Autophagy repeats information from the beginning of the paper, about the general mechanisms involved in decidualization process (lines from 962 to 993).Instead the section should be named “Decidua as a selection boundary for embryo quality”

Response 2:

As you pointed out, we deleted the earlier section and also revised the title of Section 3.

Comment 3:

In general paper is too long, with a little coherence and should be extensively rewritten.

Response 3:

We retained the scientifically important points while avoiding unnecessary repetition and shortening the manuscript as much as possible. We would be grateful if you could kindly consider the revised version for re-review.

Reviewer 3 Report

Comments and Suggestions for Authors

The paper of Shigetomi et al. is a narrative review focusing mostly but not limited to the role of apoptosis, autophagy and cellular senescence in the pathogenesis of endometriosis. Generally, one may have an impression that the authors elaborate every aspect of endometriosis pathogenicity. The paper is not original since a couple of similar reviews were published recently including IJMS. Many parts are worded in a long-winded way or duplicated. The inclusion criteria of references are unclear, so the authors are encouraged to include a Prisma flow diagram for better understanding of literature selection. The title is misleading and should be changed.

I also have some more specific comments.

Ovulation is not triggered by LH.

The authors claim that progesterone resistance may underlie lesion formation; however, this phenomenon rather characterizes eutopic endometrioid lesions and it looks it may be a secondary phenomenon.

Authors’ explanation of endometriosis-associated inflammation is not true. They claim that prostanoids and proinflammatory cytokines (IL-1, TNF) play a principal role that is only partially true. Recent findings strongly suggest that endometriosis is rather related to reparatory type 2 inflammatory response with participation of M2 macrophages, Th2 and Treg responses and predominating roles of TGF-b and IL-10. This seems to create a local suppressive milieu that may facilitate survival and growth of the endometrioid cells.

The most of the authors’ statements regarding different aspects of endometriosis pathogenicity are rather hypotheses and it is difficult to figure out which one is really related to endometriosis itself. Thus, they should clearly specify what has been really confirmed in endometriosis and what is authors’ assumption based on e.g. cancer studies.

Some figures contain duplicated information. Regarding Fig. 9 it is unclear what is the meaning of “mouse placentation” picture.

Overall, while the paper is generally interesting it should be substantially shortened with specific attention paid to what appears the most interesting e.g. senescence, autophagy and apoptosis.

Author Response

Editor-in-Chief

International Journal of Molecular Sciences (IJMS)

We would like to express our sincere gratitude to you and to the reviewers for providing insightful comments and constructive suggestions on our manuscript (Manuscript ID: ijms-3807200). We carefully considered each point raised during the review process and made substantial revisions to address the concerns. In revising the manuscript, we have endeavored to remove unnecessary repetition, condense the text where appropriate, and ensure that only the most scientifically significant aspects are emphasized.

All revisions have been marked using the track changes function in Microsoft Word, with the modified text highlighted in blue for clarity. A detailed point-by-point response to the reviewers’ comments is provided below. We believe that these revisions have considerably improved the clarity, scientific rigor, and overall quality of the manuscript.

We are truly grateful for the opportunity to resubmit our work after revision, and we sincerely hope that the revised version will now be suitable for publication in IJMS. We once again thank you for your time and consideration, and we look forward to your response.

Yours sincerely,

Hiroshi Kobayashi

E-mail: hirokoba@naramed-u.ac.jp

Point-by-point responses to reviewer comments

Reviewer 3

Comment 1:

The paper of Shigetomi et al. is a narrative review focusing mostly but not limited to the role of apoptosis, autophagy and cellular senescence in the pathogenesis of endometriosis. Generally, one may have an impression that the authors elaborate every aspect of endometriosis pathogenicity. The paper is not original since a couple of similar reviews were published recently including IJMS. Many parts are worded in a long-winded way or duplicated. The inclusion criteria of references are unclear, so the authors are encouraged to include a Prisma flow diagram for better understanding of literature selection. The title is misleading and should be changed.

Response 1:

Another reviewer raised similar concerns. Accordingly, we again ensured that only scientifically essential content was retained, redundancies were removed, and the manuscript was shortened wherever feasible. When I originally submitted the manuscript, a PRISMA flow diagram was included, but it was inadvertently omitted when the file was uploaded through the journal’s website. We have now added Figure 7.

Comment 2:

I also have some more specific comments.

Ovulation is not triggered by LH.

Response 2:

The sentence in question was deleted when the manuscript was shortened.

Comment 3:

The authors claim that progesterone resistance may underlie lesion formation; however, this phenomenon rather characterizes eutopic endometrioid lesions and it looks it may be a secondary phenomenon.

Response 3:

Thank you for your suggestion. Upon re-examining the literature, I found that Zhang et al. reported that in the eutopic endometrium of patients with endometriosis, estrogen dominance and the inflammatory milieu result in reduced expression and signaling abnormalities of the progesterone receptor, particularly PGR-B, leading to decreased progesterone responsiveness. Thus, “progesterone resistance” is now recognized as a feature not only of ectopic endometrium but also of eutopic endometrium in these patients. Therefore, we have removed the sentence you pointed out from our review.

Zhang P, Wang G. Progesterone Resistance in Endometriosis: Current Evidence and Putative Mechanisms. Int J Mol Sci. 2023 Apr 10;24(8):6992. doi: 10.3390/ijms24086992.

Comment 4:

Authors’ explanation of endometriosis-associated inflammation is not true. They claim that prostanoids and proinflammatory cytokines (IL-1, TNF) play a principal role that is only partially true. Recent findings strongly suggest that endometriosis is rather related to reparatory type 2 inflammatory response with participation of M2 macrophages, Th2 and Treg responses and predominating roles of TGF-b and IL-10. This seems to create a local suppressive milieu that may facilitate survival and growth of the endometrioid cells.

Response 4:

Thank you for your comment. We have now added explanations on M2 macrophages, Th2 cells, and Treg cells in the second paragraph of Section 2.2.

Comment 5:

The most of the authors’ statements regarding different aspects of endometriosis pathogenicity are rather hypotheses and it is difficult to figure out which one is really related to endometriosis itself. Thus, they should clearly specify what has been really confirmed in endometriosis and what is authors’ assumption based on e.g. cancer studies.

Response 5:

Thank you for your advice. During the overall shortening of the manuscript, we have carefully removed data not directly related to endometriosis.

Comment 6:

Some figures contain duplicated information. Regarding Fig. 9 it is unclear what is the meaning of “mouse placentation” picture.

Response 6:

The Figure was reduced to seven to avoid duplication. There was an error in one of the figure 6, which we have corrected.

Comment 7:

Overall, while the paper is generally interesting it should be substantially shortened with specific attention paid to what appears the most interesting e.g. senescence, autophagy and apoptosis.

Response 7:

Finally, we revised the manuscript to focus more explicitly on balancing decidualization, autophagy, and cellular senescence for reproductive success in endometriosis biology. Accordingly, we also modified the title.

Round 2

Reviewer 2 Report

Comments and Suggestions for Authors

The authors have put a lot of effort into revised manuscript and have incorporated suggestions to merge certain chapters and provided more coherent structure of the text. The manuscript is now more informative and easier to read.

There are small but important issues that still need to be addressed:

Figure 1 image needs to be corrected. Since the image represents molecular dysregulation of decidualization via FOXO1, it should shift from normal decidualization to aberrant. Instead of curved arrows pointing to panel decidualization and panel cellular senescence put lightning bolts to indicate that both processes are disturbed. Also check other figures for this issue.

Figure 6 must be altered completely. At the implantation stage the embryo is in the form of blastocyst that enters the uterus. Thus, there is no fetal blood and maternal blood contact at that stage, nor is fetal blood present at all at that stage. The initial contact is between trophoblast cells that penetrate the uterine wall to establish the placenta and the decidual cells of the mother. So, the figure of embryo acceptance/rejection needs to alter as it is false.

You mention exosomes for the first time in line 592. Either include more data in previous chapters on this issue, or avoid mentioning it here just as a statement.

I advise merging the Discussion and Future Prospects into a single chapter. Additionally, I recommend providing a more detailed Conclusion section at the end of the manuscript.

Author Response

We would like to express our sincere gratitude to both reviewers for their thorough and thoughtful evaluations, and for the considerable time they devoted to reviewing our manuscript. We have addressed all comments by revising the text, with additions and corrections highlighted in purple. Furthermore, Figures 1, 4, and 6 have been appropriately modified to reflect the suggested improvements. We hope that these revisions meet the reviewers’ expectations and that the manuscript may now be considered suitable for acceptance.

Comment 1:

The authors have put a lot of effort into revised manuscript and have incorporated suggestions to merge certain chapters and provided more coherent structure of the text. The manuscript is now more informative and easier to read.

There are small but important issues that still need to be addressed:

Figure 1 image needs to be corrected. Since the image represents molecular dysregulation of decidualization via FOXO1, it should shift from normal decidualization to aberrant. Instead of curved arrows pointing to panel decidualization and panel cellular senescence put lightning bolts to indicate that both processes are disturbed. Also check other figures for this issue.

Figure 6 must be altered completely. At the implantation stage the embryo is in the form of blastocyst that enters the uterus. Thus, there is no fetal blood and maternal blood contact at that stage, nor is fetal blood present at all at that stage. The initial contact is between trophoblast cells that penetrate the uterine wall to establish the placenta and the decidual cells of the mother. So, the figure of embryo acceptance/rejection needs to alter as it is false.

You mention exosomes for the first time in line 592. Either include more data in previous chapters on this issue, or avoid mentioning it here just as a statement.

I advise merging the Discussion and Future Prospects into a single chapter. Additionally, I recommend providing a more detailed Conclusion section at the end of the manuscript.

Response 1:

We have revised the manuscript according to your instructions. Specifically, we inserted lightning symbols in Figure 4, and we modified Figure 6 to an illustration that reflects the implantation stage. We also removed the description of exosomes. Furthermore, we merged the “Discussion” and “Future Perspectives” into a single section, and added a more detailed “Conclusion” at the end of the manuscript.

Reviewer 3 Report

Comments and Suggestions for Authors

The authors responded to the queries and corrected the manuscript along with the criticism raised. 

Author Response

Comment 1:

The authors responded to the queries and corrected the manuscript along with the criticism raised. 

Response 1:

I am deeply grateful for the thorough and thoughtful review, and for the considerable time you have dedicated to it.